# Efficient and Separate Authentication Image Steganography Network

**Junchao Zhou** [1]   **Yao Lu** [1]   **Jie Wen** [1]   **Guangming Lu** [1]

## Abstract

Image steganography hides multiple images for multiple recipients into a single cover image. All secret images are usually revealed without authentication, which reduces security among multiple recipients. It is elegant to design an authentication mechanism for isolated reception. We explore such mechanism through sufficient experiments, and uncover that additional authentication information will affect the distribution of hidden information and occupy more hiding space of the cover image. This severely decreases effectiveness and efficiency in large-capacity hiding. To overcome such a challenge, we first prove the authentication feasibility within image steganography. Then, this paper proposes an image steganography network collaborating with separate authentication and efficient scheme. Specifically, multiple pairs of lock-key are generated during hiding and revealing. Unlike traditional methods, our method has two stages to make appropriate distribution adaptation between locks and secret images, simultaneously extracting more reasonable primary information from secret images, which can release hiding space of the cover image to some extent. Furthermore, due to separate authentication, fused information can be hidden in parallel with a single network rather than traditional serial hiding with multiple networks, which can largely decrease the model size. Extensive experiments demonstrate that the proposed method achieves more secure, effective, and efficient image steganography. Code is available at https://github.com/Revive624/Authentication-Image-Steganography.

Yao Lu and Jie Wen are corresponding authors. [1]Department of Computer Science and Technology, University of Harbin Institute of Technology (Shenzhen), Guangdong, China. Correspondence to: Yao Lu <luyao2021@hit.edu.cn>, Jie Wen <jiewen_pr@126.com>.

*Proceedings of the 42nd International Conference on Machine Learning*, Vancouver, Canada. PMLR 267, 2025. Copyright 2025 by the author(s).

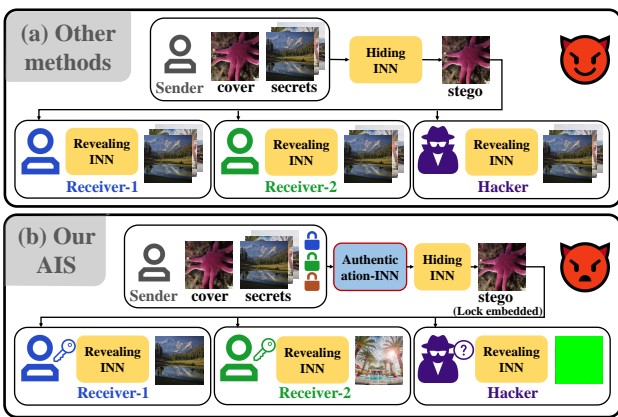

*Figure 1.* Overview of our AIS compared with traditional methods. **(a)** shows the process of other traditional methods. Secret images are revealed without certification. **(b)** shows the process of our method. Correct keys are required to reveal the secret images.

## 1. Introduction

### 1.1. Overview and problems

Image steganography involves embedding secret images within a cover image. The recipients can reveal hidden images from the stego image. For effective image steganography, the stego image should closely resemble the cover image, and recovered images must closely match the original secret images. Due to confidentiality, image steganography has been widely employed in digital watermarking (Zhang et al., 2024; Wang et al., 2023), Internet of Things (IoT) (Khari et al., 2020), military communications (Pratik et al., 2022), quantum computing (Bharatwaj & Hasabnis, 2024), healthcare (Issac & Kumar, 2023), and various domains.

Learning-based image steganography has become mainstream. Deep Neural Networks (DNNs) (Baluja, 2017; 2019; Zhang et al., 2020a) are introduced in image steganography, hiding secret images in a cover image. Some GANs-based methods (Hayes & Danezis, 2017; Chen et al., 2022; Kishore et al., 2022; Li et al., 2023) are proposed with a generator for hiding and a discriminator for identifying. Some methods (Jing et al., 2021; Lu et al., 2021; Guan et al., 2022) regard concealing and revealing as a pair of reversible processes, and introduce Invertible Neural Networks (INNs) to hide and reveal images with shared parameters. Recently,

diffusion-based models (Yu et al., 2024; Yang et al., 2024) have provided secure and robust steganography. However, current methods still suffer from three major defects.

**(1) Lack of authentication.** As shown in Figure 1, all secret images are revealed at once, without verifying the identity of recipients. This defect among multiple recipients can lead to unauthorized access and serious information leakage, which fails to meet practical application requirements.

**(2) Low quality when hiding multiple images.** Large-capacity hiding methods (Lu et al., 2021; Guan et al., 2022) hide all secret images in the limited space of a cover image. When the space is insufficient to accommodate more secret information, the network may either sacrifice the space of the cover image or lose secret information, both reducing transmission effectiveness.

**(3) Large increase in model size and computational cost.** For traditional methods of serial hiding (Guan et al., 2022; Zhou et al., 2024), multiple networks are required to be assembled to hide secret images. This results in a linear growth in model size and computational cost as the number of secret images increases.

### 1.2. Exploration and Challenges

In order to address non-authentication and achieve isolated reception, it is elegant to introduce an authentication mechanism. Following IIS (Zhou et al., 2024), we establish an **authentication-based** image steganography network. Through embedding additional lock information in the cover image, a corresponding key is required to recover the secret images. Furthermore, an **authentication-free** network is built, which removes authentication while preserving the hiding network. Based on the above two methods, we conduct comparison experiments to evaluate the effectiveness of authentication and its influence on hiding process on DIV2K dataset.

In such comparison, through a large number of samples and lots of statistics, Jensen-Shannon (JS) divergence and Peak Signal-to-Noise Ratio (PSNR) are calculated to quantify the global and local differences between cover and stego images generated by authentication-based method (blue) and authentication-free variant (red), as shown in Figure 2. From **(a)**, blue spots show a general distribution of higher overall JS divergence and lower PSNR. This indicates that authentication-based method may cause more significant alterations in the global distribution and also reduce the local similarity to the cover images. From **(b)**, authentication-based method shows a great decline in PSNR and SSIM on average, indicating that the additional lock information will decrease the similarity between cover and stego images.

Based on the insights gained from the results, our exploration of authentication mechanism highlights *two crit-*

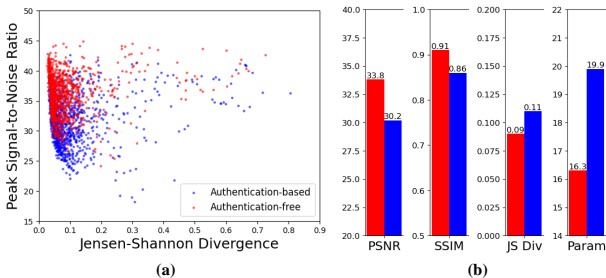

*Figure 2.* **(a)** Comparisons of JS Divergence and PSNR between cover and stego images of authentication-free and authentication-based methods. **(b)** Comparisons of PSNR, SSIM, JS Divergence and model parameters between authentication-free and authentication-based methods on average.

*ical challenges* for building authentication-based image steganography, which further motivates our improvement on embedding lock information.

**(1) Embedding authentication information while maintaining the quality of stego and revealed secret images.** Due to the inconsistent distribution of secret images and locks, the embedded information usually occupies considerable hiding space of the cover image, resulting in a great drop in the quality of both stego and revealed secret images and reducing the transmission effectiveness.

**(2) Integrating authentication while limiting the model size.** Compared to authentication-free method, the parameter number of authentication-based method grows from 16.3M to 19.9M, shown in Figure 2 **(b)**. The extra parameters bring additional computational cost and severely reduce transmission efficiency.

### 1.3. Solution

To overcome the challenges of existing methods, we propose an Efficient and Separate Authentication Image Steganography Network (AIS) with two collaborative stages, consisting of an Invertible Authentication Network (IAN) and an Invertible Hiding Network (IHN), respectively.

**In the first stage**, IAN introduces an authentication mechanism. The mechanism embeds locks in secret images to ensure that only recipients with correct keys can access the hidden secret images. Locks and keys are generated from a dynamic lock-key generation strategy by learning features of both the cover and secret images. What is more, different from traditional one-stage methods, our IAN fuses the locks and secret images. Then, such fused information will be embedded in the cover image in the second stage (IHN). This will produce a better distribution adaptation and may free up space for hiding information. Additionally, IAN extracts more reasonable primary information from secret images. This reduces the secret information to be hidden.

All above strategies can both achieve isolated reception and produce high-quality stego and revealed secret images.

**In the second stage**, IHN utilizes the fused lock and secret information with better distribution adaptation in the hiding process to produce high-quality stego images. Furthermore, due to separate authentication, IHN can hide all secret images in parallel. This approach only requires training a single hiding network, unlike serial hiding methods that require an extra network for each secret image. The main contributions of our method are summarized as follows:

- We propose an Authentication Invertible Steganography (AIS) collaboratively consisting of an Invertible Authentication Network (IAN) and an Invertible Hiding Network (IHN). The network enables isolated reception and enhances security among multi-recipients.

- The proposed IAN can generate pairs of lock-key and fuse the locks and secret images to make distribution adaptation. This will decrease the locks occupation of the hiding space in cover images.

- The proposed IAN also contains learnable mappings to extract more reasonable primary information from secret images, producing high-quality stego and revealed secret images on the basis of isolated reception.

- The proposed IHN employs the fused secret and lock information to produce stego images. IHN also allows for efficient parallel hiding of secret images, requiring only a single network to be trained, significantly reducing the model size and computational cost.

## 2. Related Work

### 2.1. Authentication-Free Image Steganography

Image steganography aims to conceal secret images within a single cover image, allowing recipients to accurately reveal the hidden information. Baluja (Baluja, 2017; 2019) proposed the first deep neural network for image steganography. HiNet (Jing et al., 2021) used INNs for reversible hiding and revealing in frequency domain. ISN (Lu et al., 2021) extended INNs to parallel processing in spatial domain for multiple secret images. DeepMIH (Guan et al., 2022) explored a serial hiding strategy with an importance map for optimized spatial utilization. InvMIHNet (Chen et al., 2024) achieved large-capacity hiding by splicing multiple images into a single secret image. Recently, CRoSS (Yu et al., 2024) explored DDIM Inversion to transform secret images into stego images without a cover image. However, these methods do not support recipient verification and reveal all secret images simultaneously. This increases the risk of unauthorized access and reduces practical flexibility.

### 2.2. Authentication-Based Image Steganography

Image steganography with certification is challenging, as it requires embedding authentication information while minimizing the impact on stego and revealed secret images. Kweon et al. (Kweon et al., 2021) proposed a key mechanism based on an encoder-decoder structure. IIS (Zhou et al., 2024) dynamically generated global and local keys for each image, embedding fused keys into stego images via an INN structure for authentication. DiffStega (Yang et al., 2024) achieved coverless steganography through diffusion models, where an image prompt serves as a private key guiding the revealing. Although these methods enable authentication, the authentication information often degrades the quality of hiding and recovery, reducing transmission effectiveness. Furthermore, these methods typically have large model sizes, struggling to balance verifiability and efficiency in large-capacity hiding.

### 2.3. Invertible Neural Networks

Invertible Neural Networks (INNs) establish a bijective mapping between data distribution $p_x$ and latent distribution $p_z$. Dinh et al. (Dinh et al., 2015; 2017) introduced coupling layers in generative flow models. Gilbert et al. (Gilbert et al., 2017) explored the reversibility of INNs. Kingma et al. (Kingma & Dhariwal, 2018) improved generative flow models using invertible $1 \times 1$ convolutional layers. Xiao et al. (Xiao & Liu, 2020) proposed matrix exponential coupling layers for improved density estimation performance. Ardizzone et al. (Ardizzone et al., 2019) proposed Conditional Invertible Neural Networks (CINNs), where a condition guides the generation. Koehler et al. (Koehler et al., 2021) theoretically explored the depth and condition of normalizing flows, which highlights the trade-off between authentication and generation quality in image steganography.

To address these challenges, we propose distribution adaptation and secret information extraction. This enables isolated reception while ensuring high quality for both stego and revealed secret images. Separate authentication promotes parallel hiding, thereby reducing the model size.

## 3. Method

In this part, we first demonstrate the feasibility of incorporating authentication in our two-stage image steganography. Then, we detail the structure of our proposed Efficient and Separate Authentication Image Steganography Network (AIS), consisting of an Invertible Authentication Network (IAN) and an Invertible Hiding Network (IHN), as shown in Figure 3. IAN embeds locks in secret images and verifies keys during the revealing process, while extracting more reasonable primary information of secret images. IHN hides extracted information in the cover image and reveals it from

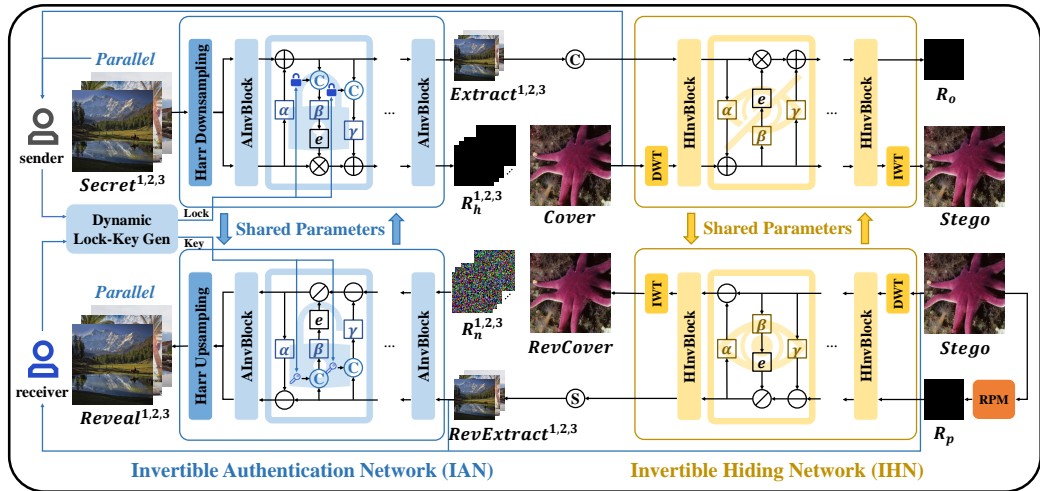

*Figure 3.* The detail structure of Invertible Authentication Network (IAN) and Invertible Hiding Network (IHN). $\otimes$ represents Hadamard product. $\oplus$ represents element-wise addition. $\copyright$ represents channel concatenation. $\circledS$ represents channel split.

the stego image. Locks and keys for IAN are generated by a dynamic generation module, a simplified UNet structure (Ronneberger et al., 2015). Further details are provided in Appendix B.

### 3.1. Feasibility of Authentication

Image steganography involves hiding and revealing processes, which can be modeled as a normalizing flow $f$ (Dinh et al., 2015). $x$ denotes the data domain of secret images and $z$ denotes the latent domain. The probability distribution $p_x(x)$ of the secret images can be derived via the change-of-variable formula:

$$\hat{p}_x(x;\theta) = p_z(f_\theta(x)) \cdot \left| \det\left( \frac{\partial f_\theta(x)}{\partial x} \right) \right|, \quad (1)$$

where $det(\cdot)$ denotes the determinant. Given data samples $\{x^i\}_{i=1}^N$, the parameters $\theta$ can be optimized by maximizing log likelihood loss:

$$L_\theta = \sum_{i=1}^{N} \left( log(p_z(f_\theta(x^i))) + log \left| \det\left( \frac{\partial f_\theta(x^i)}{\partial x^i} \right) \right| \right). \quad (2)$$

To incorporate authentication information, we adopt the conditional normalizing flow framework (Ardizzone et al., 2019; 2021; Wen et al., 2023). The lock and key are denoted by $c$ and $c'$, respectively. The forward mapping is defined as: $z = f_\theta(x, c)$ and the backward mapping is defined as: $z = f_\theta(x, c')$. The change-of-variable formula is:

$$\hat{p}_{x|c}(x|c;\theta) = p_z(f_\theta(x, c)) \cdot \left| \det\left( \frac{\partial f_\theta(x, c)}{\partial x} \right) \right|. \quad (3)$$

Through the flow, the distribution of the secret images is influenced by the authentication information of the lock.

Through training, $\hat{x}$ can be shifted away from the distribution of $x$ when given the wrong key, while $\hat{x}$ can approximate the distribution of $x$ when given the correct key. A detailed proof is provided in Appendix A.2.

### 3.2. Overall framework

**In the forward hiding process**, IAN first generates locks with the dynamic lock-key generation module. Then, locks and secret images are fused to make distribution adaptation. Meanwhile, IAN extracts more reasonable primary information from the secret images. The cover image is transformed into the frequency domain to match the extracted information in size. Both the cover image and extracted information are fed into the IHN to produce the stego image, while redundant information is omitted during transmission. **In the backward revealing process**, following IIS (Zhou et al., 2024), redundancy is restored using a Redundancy Prediction Module to better fit the latent distribution. The primary information is recovered through the reverse process of IHN. Then, a key is generated by the dynamic lock-key generation module, and used in the IAN for verification, gradually decoupling the complete secret image.

### 3.3. Invertible Authentication Network

IAN comprises several authentication invertible blocks, a Haar Downsampling Module and a Haar Upsampling Module. To enable isolated reception, each secret image is processed independently rather than as a combined input.

**In the forward hiding process**, the Haar Downsampling Module transforms a RGB secret image $S_o \in \mathcal{R}^{B \times 3 \times H \times W}$ from the spatial domain to the frequency domain $S_f \in \mathcal{R}^{B \times 12 \times (H/2) \times (W/2)}$. $S_f$ is then decomposed into a

low-frequency component $S_l \in \mathcal{R}^{B \times 3 \times (H/2) \times (W/2)}$ and a high-frequency component $S_h \in \mathcal{R}^{B \times 9 \times (H/2) \times (W/2)}$, which are fed into the authentication invertible blocks. The transformation in the $i^{th}$ block is described by:

$$x^{i+1} = x^i + \alpha(y^i),$$
$$y^{i+1} = y^i \otimes \exp(\beta(x^{i+1}, L)) + \gamma(x^{i+1}, L), \quad (4)$$

where $x$ and $y$ represent the low-frequency and high-frequency components, $L$ denotes the lock, $\otimes$ is Hadamard product, $\exp(\cdot)$ is exponential function. The functions $\alpha(\cdot)$, $\beta(\cdot)$ and $\gamma(\cdot)$ are learnable mappings based on DenseNet structures (Huang et al., 2017). Unlike general INNs, IAN concatenates $x^{i+1}$ and the lock $L$ along the channel dimension as inputs to these mappings. Through these learnable mappings, IAN makes distribution adaptation between locks and secret images, outputting the extracted information $S_c$ with 3 channels and redundant information $R_h$.

**In the backward revealing process**, since $R_h$ in the forward process is omitted, an auxiliary variable is introduced. Following (Xiao et al., 2023), a latent variable $R_n \in \mathcal{R}^{B \times 9 \times (H/2) \times (W/2)}$ is sampled from a Gaussian distribution. The invertible blocks iteratively recover the secret images from this latent variable under the key $K$ for verification. The reverse transformation is given by:

$$y^i = (y^{i+1} - \gamma(x^{i+1}, K)) \otimes \exp(-\beta(x^{i+1}, K)),$$
$$x^i = x^{i+1} - \alpha(y^i), \quad (5)$$

where $x$ represents the revealed extracted information, $y$ denotes the auxiliary latent variable. The parameters of $\alpha(\cdot)$, $\beta(\cdot)$ and $\gamma(\cdot)$ are shared between the forward and backward processes. The key $K$ must closely match the embedded lock $L$ for accurate decoupling of the secret image. Otherwise, deviations in the key will introduce errors, causing the revealed secret image to diverge from the original secret image. Given the correct key, the revealed secret image $S_{ro}$ is reconstructed using the Haar Upsampling Module.

### 3.4. Invertible Hiding Network

The Invertible Hiding Network (IHN) consists of several hiding invertible blocks, a Discrete Wavelet Transformation (DWT) module and an Inverse Discrete Wavelet Transformation (IWT).

**In the forward hiding process**, the RGB cover image $C_o \in \mathcal{R}^{B \times 3 \times H \times W}$ is transformed from the spatial domain to the frequency domain $C_f \in \mathcal{R}^{B \times 12 \times (H/2) \times (W/2)}$ via DWT. This transformation not only aligns $C_f$ with $S_c$ in dimensions, but also enhances hiding quality and robustness against steganalysis (Guan et al., 2022). Thanks to separate authentication in IAN, IHN can hide secret images in parallel. The secret image components $S_c^{1,2,3}$ are concatenated along the channel dimension and, together with $C_f$,

are fed into the invertible blocks. The computation within the $i^{th}$ invertible block is described as:

$$x^{i+1} = x^i + \alpha(y^i),$$
$$y^{i+1} = y^i \otimes \exp(\beta(x^{i+1})) + \gamma(x^{i+1}), \quad (6)$$

where $x$ and $y$ represent the cover and secret information, respectively. The functions $\alpha(\cdot)$, $\beta(\cdot)$ and $\gamma(\cdot)$ share the same structure as those in the IAN. The forward process produces a stego image $T_f$ and redundant information $R_o$, and $T_f$ is restored to the spatial domain $T$ via IWT.

**In the backward revealing process**, following (Zhou et al., 2024), a Redundancy Prediction Module (RPM) is employed to ensure reversibility and enhance the quality of revealed secret images. RPM is a learnable module with a residual structure as described in (Mou et al., 2023). It adapts an auxiliary variable $R_p$ from the stego image $T$ to closely approximate $R_o$. This ensures the reversibility of the invertible blocks and preserves high fidelity in the revealed secret images. Using $R_p$ and the frequency-domain stego image $T_f$, the reverse computation is defined as:

$$y^i = (y^{i+1} - \gamma(x^{i+1})) \otimes \exp(-\beta(x^{i+1})),$$
$$x^i = x^{i+1} - \alpha(y^i), \quad (7)$$

where the invertible blocks gradually decouple the extracted information from the stego image. Channel split outputs the revealed extracted information $S_{rc}$.

### 3.5. Loss Functions

Stego images should closely resemble the cover images. The hiding loss is defined as:

$$L_h = \mathcal{L}_2(C_o, T) + \mathcal{L}_2(C_l, T_l), \quad (8)$$

where $\mathcal{L}_2$ represents the Mean Square Error (MSE) loss, $T_l$ and $C_l$ are the low-frequency components of $T$ and $C_o$ after DWT. The low-frequency loss part enhances visual similarity and improves resistance to steganalysis (Guan et al., 2022).

Revealed secret images should closely match the original secret images for effective transmission. Thus, the revealing loss consists of:

$$L_{rc} = \sum_{i=1}^{N} \mathcal{L}_2(S_c^i, S_{rc}^i),$$
$$L_r = \sum_{i=1}^{N} \mathcal{L}_2(S_o^i, S_{ro}^i) + \sum_{i=1}^{N} \mathcal{JS}(S_o^i, S_{ro}^i), \quad (9)$$

where $\mathcal{JS}$ denotes Jensen-Shannon Divergence loss, which further improves the quality of $S_{ro}$ (Chen et al., 2024).

*Table 1.* Comparisons between our method and baselines hiding 2, 3, 4, 5 secret images, on DIV2K dataset and ImageNet dataset.

| N | METHOD | PARAMS | FLOPs | TIME | DIV2K | | | | | | IMAGENET | | | | | |
|---|---|---|---|---|---|---|---|---|---|---|---|---|---|---|---|---|
| | | | | | COVER-STEGO | | | SECRET-REVEAL | | | COVER-STEGO | | | SECRET-REVEAL | | |
| | | | | | PSNR↑ | SSIM↑ | LPIPS↓ | PSNR↑ | SSIM↑ | LPIPS↓ | PSNR↑ | SSIM↑ | LPIPS↓ | PSNR↑ | SSIM↑ | LPIPS↓ |
| 2 | ISN | 3.17M | 414.1G | 46.0ms | 34.661 | 0.845 | 0.502 | 33.734 | 0.858 | 0.474 | 34.489 | 0.833 | 0.358 | 33.470 | 0.835 | 0.516 |
| | DEEPMIH | 12.42M | 426.5G | 103.4ms | 37.460 | 0.871 | 0.209 | 35.969 | 0.910 | 0.206 | 37.209 | 0.863 | 0.364 | 33.723 | 0.885 | 0.736 |
| | IIS | 22.30M | 718.0G | 180.4ms | 34.619 | 0.845 | 0.592 | 37.471 | 0.909 | 0.236 | 34.751 | 0.854 | 0.528 | 38.106 | 0.900 | 0.278 |
| | **AIS(OURS)** | 5.57M | 186.0G | 69.8ms | 42.141 | 0.913 | 0.130 | 38.088 | 0.944 | 0.201 | 41.345 | 0.917 | 0.176 | 35.802 | 0.911 | 0.455 |
| 3 | ISN | 3.34M | 436.4G | 49.41ms | 31.233 | 0.850 | 0.564 | 30.049 | 0.840 | 1.136 | 33.525 | 0.826 | 0.576 | 31.849 | 0.811 | 0.891 |
| | DEEPMIH | 19.44M | 676.4G | 157.1ms | 31.286 | 0.768 | 1.089 | 27.298 | 0.832 | 2.260 | 33.995 | 0.819 | 0.365 | 29.560 | 0.837 | 0.949 |
| | IIS | 33.40M | 1076.4G | 264.6ms | 30.395 | 0.799 | 1.030 | 33.347 | 0.850 | 0.537 | 28.299 | 0.725 | 1.515 | 31.249 | 0.804 | 0.821 |
| | **AIS(OURS)** | 5.76M | 192.1G | 87.5ms | 34.721 | 0.904 | 0.396 | 34.761 | 0.903 | 0.420 | 33.334 | 0.849 | 0.511 | 32.033 | 0.854 | 0.790 |
| 4 | ISN | 3.51M | 458.6G | 50.4ms | 29.488 | 0.712 | 1.148 | 29.862 | 0.786 | 1.251 | 32.299 | 0.816 | 0.618 | 30.264 | 0.807 | 1.093 |
| | DEEPMIH | 26.46M | 926.3G | 218.3ms | 28.978 | 0.682 | 1.438 | 23.581 | 0.740 | 4.702 | 31.286 | 0.776 | 0.948 | 26.068 | 0.768 | 3.062 |
| | IIS | 44.51M | 1434.7G | 344.8ms | 27.121 | 0.746 | 4.107 | 29.970 | 0.841 | 3.230 | 27.708 | 0.699 | 1.929 | 27.820 | 0.769 | 2.314 |
| | **AIS(OURS)** | 5.95M | 198.2G | 102.3ms | 34.947 | 0.887 | 0.608 | 34.100 | 0.908 | 0.591 | 31.711 | 0.863 | 0.915 | 31.048 | 0.841 | 0.901 |
| 5 | ISN | 3.68M | 480.9G | 51.7ms | 26.735 | 0.650 | 2.443 | 27.374 | 0.713 | 2.366 | 30.522 | 0.777 | 1.018 | 29.077 | 0.790 | 1.544 |
| | DEEPMIH | 33.48M | 1176.1G | 269.1ms | 29.477 | 0.692 | 1.594 | 22.189 | 0.716 | 5.880 | 32.507 | 0.792 | 0.815 | 26.308 | 0.786 | 3.069 |
| | IIS | 55.62M | 1793.0G | 439.6ms | 26.676 | 0.741 | 3.194 | 28.676 | 0.825 | 2.147 | 24.345 | 0.650 | 3.927 | 26.293 | 0.747 | 2.817 |
| | **AIS(OURS)** | 6.15M | 204.3G | 121.1ms | 36.149 | 0.887 | 0.334 | 30.765 | 0.854 | 1.141 | 32.767 | 0.841 | 0.585 | 30.060 | 0.833 | 1.055 |

Due to strict authentication, keys should closely match the generated locks. This is represented by the key loss:

$$L_k = \sum_{i=1}^{N} \mathcal{L}_1(L^i, K^i), \qquad (10)$$

where $\mathcal{L}_1$ represents the Mean Absolute Error (MAE) loss.

The authentication mechanism requires that entering a wrong key should cause the revealed information to diverge from the secret image. This is enforced by the triplet loss:

$$L_t = \sum_{i=1}^{N} max\{0, \mathcal{L}_2(S_o^i, S_{ro}^i) - \mathcal{L}_2(S_o^i, S_{rn}^i) + \text{margin}\}, \qquad (11)$$

where $S_{rn}$ is a negative sample generated using a random key during the revealing process, and margin is a constant set to 1 (Schroff et al., 2015).

Following literature (Zhou et al., 2024), a redundancy loss is defined to ensure similarity between the predicted redundancy and the forward output of the redundant information:

$$L_p = \sum_{i=1}^{N} \mathcal{L}_2(R_o^i, R_p^i). \qquad (12)$$

The overall training loss is the weighted sum of the above losses:

$$L = \lambda_1 L_h + \lambda_2 L_{rc} + \lambda_3 L_r + L_k + L_t + L_p, \qquad (13)$$

where $\lambda_1$, $\lambda_2$, $\lambda_3$ are hyper-parameters that balance the contribution of each loss term.

# 4. Experiments

We compare our method with three baselines: ISN (Lu et al., 2021), DeepMIH (Guan et al., 2022) and IIS (Zhou

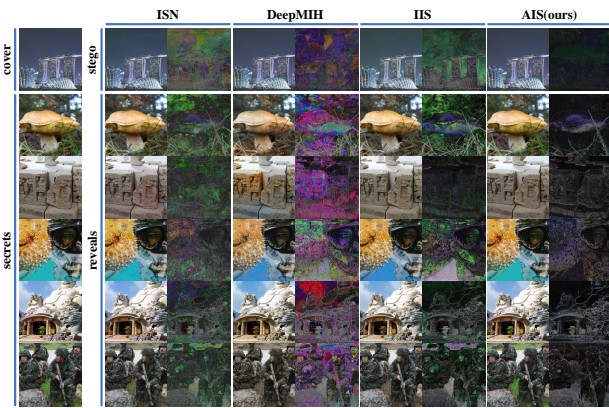

*Figure 4.* The visual results of experiment of our AIS method and baselines on DIV2K dataset, hiding 5 secret images.

et al., 2024) in terms of PSNR, SSIM and LPIPS, on DIV2K dataset and ImageNet dataset. The training settings and details are provided in Appendix C.

## 4.1. Quality Analysis

**Quantitative Results.** Table 1 presents quantitative comparisons. The results for the revealed secret images are reported as averages. For clarity, LPIPS values are scaled by $10^3$. **On DIV2K dataset**, our AIS achieves superior performance across all metrics. For stego images, PSNR, SSIM, and LPIPS show improvements of **5.062 dB**, **0.096**, and **0.512**, respectively. For the revealed secret images, PSNR, SSIM, and LPIPS are optimized by **2.063 dB**, **0.046**, and **0.047**, respectively. **On ImageNet dataset**, our AIS achieves SSIM improvements of **0.043** and **0.026** for stego images and revealed secret images, respectively. Similar enhancements are observed in PSNR and LPIPS.

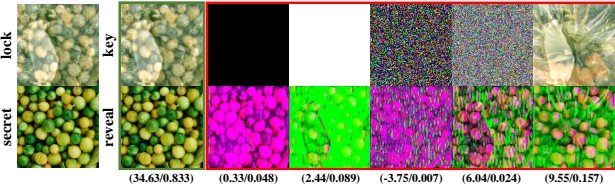

*Figure 5.* The visual results of experiments on the effectiveness of the authentication mechanism. Different forms of wrong keys are used to verify the uniqueness of the correct key.

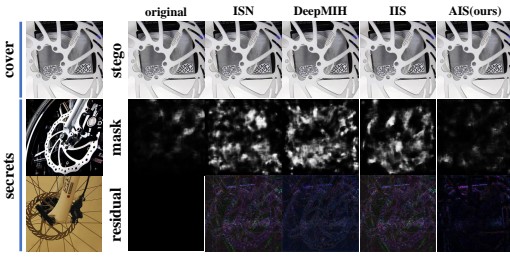

*Figure 6.* Visual results on ManTraNet among baselines and our AIS method.

*Table 2.* Detection accuracy on SRNet and ZhuNet among baselines and our AIS method.

| METHODS | ACCURACY (%) | | |
|---|---|---|---|
| | SRNET | ZHUNET | LWENET |
| ISN | 73.50 | 67.80 | 67.30 |
| DEEPMIH | 60.95 | 74.00 | 89.55 |
| IIS | 61.70 | 81.75 | 62.20 |
| AIS(OURS) | 52.80 (8.15%↓) | 61.40(6.40%↓) | 51.75(10.45%↓) |

**Qualitative Results.** Figure 4 presents the visual comparisons between our method and other baselines. Residual errors, amplified by a factor of 10, are used to emphasize differences between the generated images and the ground-truth images. From the results, significant color distortions appear in the stego images and revealed secret images from other methods. This indicates that a single cover image cannot effectively accommodate information from multiple secret images, leading to color loss. In contrast, our AIS method, which hides only extracted information, narrows the gap between the stego and cover images while preserving the fine details of the secret images.

The above results demonstrate that our AIS method significantly improves both the stego and revealed secret images. Our method effectively extracts more reasonable primary features of secret images, reducing the information needed for hiding and revealing. Meanwhile, our AIS integrates the lock with the secret images, partially avoiding embedding information that has an inconsistent distribution in the cover image. This may reduce the occupation of hidden space. Both allow a single cover image to accommodate more secret images, while preserving sufficient information for high-quality restoration.

### 4.2. Security Analysis

#### 4.2.1. EFFECTIVENESS OF AUTHENTICATION

To demonstrate the effectiveness of our proposed authentication mechanism, we simulate attempts to reveal secret images using random keys on DIV2K dataset. As shown in Figure 5, we use incorrect keys, such as all zeros, all ones, Gaussian distribution, uniform distribution, and a key of another secret image. The restored images are almost indistinguishable. Clearly, information revealed using randomly

generated keys is almost meaningless among unauthorized recipients.

These results imply that the authentication mechanism effectively identifies recipients. Secret images are restored independently with correct keys. When using a forged key, the revealed secret images become severely corrupted. This enhances the system's security against attacks and provides flexibility in sending different images to different recipients.

#### 4.2.2. RESISTANCE AGAINST STEGANALYSIS

Steganalysis methods aim to detect hidden information in stego images. A robust steganography method must perform well in both quality and resistance to steganalysis. Otherwise, attackers can easily identify stego images, compromising transmission confidentiality and integrity.

**Detection of Stego Image.** We test our method against three state-of-the-art steganalysis methods, SRNet (Boroumand et al., 2019), ZhuNet (Zhang et al., 2020b), and LWENet (Weng et al., 2022). 1000 cover-stego image pairs are generated using baselines and our AIS on ImageNet dataset. Table 2 shows that our AIS achieves the lowest detection accuracy among all methods, decreasing by **8.15%**, **6.40%** and **10.45%**. This improvement is due to the proposed IAN in our method, which reduces the amount of hidden information, making stego and cover images harder to distinguish.

**Detection of Secret Information.** To evaluate potential information leakage, we employ ManTraNet (Wu et al., 2019), a method designed to detect the locations of hidden data within an image. The white regions in the mask highlight detected anomalies. Figure 6 shows fewer abnormal areas detected by ManTraNet in stego images of our method. While the white areas detected in other methods reveal contours of hidden secret images, those detected in our AIS hardly reveal any meaningful information. These results demonstrate superior security of our method in preventing information leakage.

*Table 3.* Ablation experiments on the channel number of extracted information in the proposed IAN.

| CHANNELS | COVER-STEGO | | | SECRET-REVEAL | | |
|---|---|---|---|---|---|---|
| | PSNR↑ | SSIM↑ | LPIPS↓ | PSNR↑ | SSIM↑ | LPIPS↓ |
| 12 | 33.488 | 0.876 | 0.639 | 27.846 | 0.790 | 2.542 |
| 9 | 30.568 | 0.845 | 1.232 | 27.751 | 0.819 | 3.079 |
| 6 | 29.472 | 0.798 | 1.548 | 26.767 | 0.791 | 3.094 |
| **3(OURS)** | **36.149** | **0.887** | **0.334** | **30.765** | **0.854** | **1.141** |

## 4.3. Efficiency Analysis

Table 1 summarizes parameter number, Floating-point Operations (FLOPs), and inference time. Across all settings of $N$, our AIS uses fewer parameters than IIS and DeepMIH. Notably, when $N = 5$, AIS requires only **6.15M** parameters, far fewer than IIS (**55.62M**) and DeepMIH (**33.48M**). In terms of FLOPs, our AIS shows superior computational efficiency, saving an average of **252.35G** compared to ISN. Additionally, our method significantly reduces inference time compared to DeepMIH and IIS. Due to the superior design of AIS, parallel hiding significantly reduces model size and computational cost. As the number of secret images increases, our method retains its efficiency advantage. This scalability demonstrates its suitability for large-capacity steganography applications.

## 4.4. Ablation Studies

### 4.4.1. EXTRACTION CHANNELS

The proposed IAN extracts features of 3 channels from secret images, focusing on retaining the primary information. To assess the performance among different channel numbers, we conduct ablation experiments on DIV2K dataset. The results summarized in Table 3 show that 3 channels yield the best quality for both stego and revealed secret images. This indicates that increasing the channel number intensifies recovery errors caused by the proposed IHN, leading to significant information loss and degraded image quality in

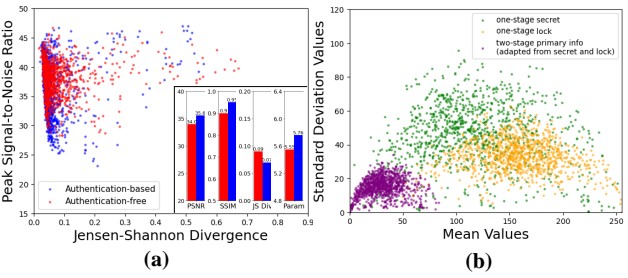

*Figure 7.* **(a)** Comparisons of PSNR, SSIM, JS Divergence and model parameters between authentication-free method and our AIS. **(b)** Comparisons of mean and standard deviation between secret images and locks in one-stage method and extracted information in our method.

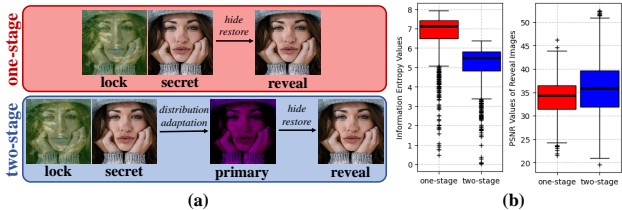

*Figure 8.* **(a)** Example of hiding process of one-stage method and our two-stage AIS method. **(b)** Comparisons of information entropy of hidden information and PSNR of revealed secret images between one-stage method and our two-stage AIS method.

scenarios with more channels.

### 4.4.2. EFFECTIVENESS OF DISTRIBUTION ADAPTATION

To demonstrate the IAN's effectiveness of distribution adaptation in our AIS method (blue), we compare it with an authentication-free variant (red) modified from AIS and a one-stage authentication method on DIV2K dataset. As illustrated in Figure 7 **(a)**, our method achieves a lower overall JS Divergence and higher PSNR and SSIM. This suggests that distribution adaptation partially avoids embedding information with inconsistent distribution in the cover image. Moreover, due to parallel hiding, only a small number of parameters has been increased. **(b)** shows the mean and standard deviation to compare the information distribution between the one-stage method and our two-stage AIS method. In one-stage method, the distribution of locks (orange) and secret images (green) is inconsistent, whereas our AIS (purple) achieves a more consistent distribution. This indicates that the proposed IAN adapts secret images and locks to fused information with a consistent latent distribution. Through such adaptation, AIS may avoid the additional lock information to occupy the hiding space, enhancing the quality of both stego and revealed secret images.

### 4.4.3. EFFECTIVENESS OF PRIMARY INFORMATION

To evaluate the effectiveness of the primary information extracted by the proposed IAN, we compare our method with one-stage method without extraction on DIV2K dataset. **(a)** shows the difference of the hiding and revealing processes between one-stage method and our two-stage AIS. This instance displays that the secret image can be restored from the primary information. In **(b)**, a boxplot compares information entropy between extracted information (blue) and secret images (red), and another boxplot compares PSNR of revealed secret images between one-stage method (red) and our AIS (blue). Compared to one-stage method, our AIS decreases the average information entropy by **1.85**, but improves the average PSNR by **2.086 db**. The results indicates that our method may filter the redundant information to some extent, and contains less redundant information, extracting more reasonable primary information while pre-

serving the quality of the revealed secret images.

## 5. Conclusion

This paper demonstrates the feasibility of an authentication mechanism and proposes an Efficient and Separate Authentication Image Steganography (AIS) method. AIS embeds locks in secret images. Distribution adaptation partially releases the space in cover images and extracts primary information for hiding, enhancing the quality of both stego and secret images. A correct key is needed to recover the secret image, achieving isolated reception among different recipients. Due to separate authentication, only a single network needs to be trained, significantly limiting the model size. The specially designed two-stage method enables secure, effective, efficient, and flexible image steganography.

## Impact Statement

This paper presents work whose goal is to advance the field of Machine Learning. There are many potential societal consequences of our work, none which we feel must be specifically highlighted here.

## Acknowledgments

This work was supported in part by the NSFC fund (NO. 62206073, 62176077), in part by the Shenzhen Key Technical Project (NO. JSGG20220831092805009, JSGG20220831105603006, JSGG20201103153802006, KJZD20230923115117033, KJZD20240903100712017), in part by the Guangdong International Science and Technology Cooperation Project (NO. 2023A0505050108), in part by the Shenzhen Fundamental Research Fund (NO. JCYJ20210324132210025), and in part by the Guangdong Provincial Key Laboratory of Novel Security Intelligence Technologies (NO. 2022B1212010005), and in part by the Natural Science Foundation of Shenzhen General Project under Grant JCYJ20240813110007010, in part by the Natural Science Foundation of Guangdong Province under Grant 2023A1515010893, in part by the Shenzhen Doctoral Initiation Technology Plan under Grant RCBS20221008093222010, in part by the Shenzhen Pengcheng Peacock Startup Fund.

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

# A. Derivations

### A.1. Derivation of Equation (2)

In Maximum Likelihood Estimation (MLE), the likelihood function $L(\theta|x)$ can be calculated as:

$$L(\theta|x) = p(x|\theta). \tag{14}$$

Taking the logarithm on both sides, we have the log likelihood loss $L_\theta$:

$$
\begin{aligned}
L_\theta &= log(L(\theta|x)) \\
&= log(\hat{p}(x|\theta)) \\
&= log \prod_{i=1}^{N} \hat{p}(x^i|\theta) \\
&= \sum_{i=1}^{N} log(\hat{p}(x^i|\theta)) \\
&= \sum_{i=1}^{N} \left( log(p_z(f_\theta(x^i))) + log \left| det \left( \frac{\partial f_\theta(x^i)}{\partial x^i} \right) \right| \right).
\end{aligned}
\tag{15}
$$

For KL Divergence, we have:

$$
\begin{aligned}
D_{KL}(p_x||\hat{p}_x) &= \int p_x(x) log \frac{p_x(x)}{\hat{p}_x(x;\theta)} dx \\
&= \int p_x(x) log(p_x(x)) dx - \int p_x(x) log(\hat{p}_x(x;\theta)) dx.
\end{aligned}
\tag{16}
$$

As the former part is constant, minimizing KL Divergence is equivalent to maximizing the latter part. We use samples $\{x^i\}_{i=1}^{N} \sim p_x(x)$ to estimate the expectation of this part of the formula:

$$L_\theta = \frac{1}{N} \sum_{i=1}^{N} \log \hat{p}_x(x^i; \theta). \tag{17}$$

Omitting $\frac{1}{N}$ and with Equation (1), we have:

$$L_\theta = \sum_{i=1}^{N} \left( log(p_z(f_\theta(x^i))) + log \left| det \left( \frac{\partial f_\theta(x^i)}{\partial x^i} \right) \right| \right). \tag{18}$$

The above derivation proves that maximizing $L_\theta$ is equivalent to minimizing KL Divergence, aligning with the goal of approximating $\hat{p}_x(x)$ with $p_x(x)$.

### A.2. Validation of Condition $c$ as Key

In this part, we try to prove the feasibility of adding a condition $c$ as the key for verification. The process can be formulated as $z = f_\theta(x, c)$ and $\hat{x} = f_\theta^{-1}(z, c')$. This implies two requirements: **(a)** $\hat{x} \sim p_x(x)$ when $c'$ is consistent to $c$. **(b)** Maximizing the distance between $\hat{x}$ and $x$ when $c'$ is inconsistent to $c$. We start from Equation (3):

$$\hat{p}_{x|c'}(x|c';\theta) = p_{z|c'}(f_\theta(x,c)|c') \cdot \left| det \left( \frac{\partial f_\theta(x,c)}{\partial x} \right) \right|. \tag{19}$$

**Case (a)**: With $c' \equiv c$, we have:

$$
\begin{aligned}
\hat{p}_x(x;\theta) &= \int \hat{p}_{x|c'}(x|c';\theta)dc' \\
&= \int p_{z|c}(f_\theta(x,c)|c) \cdot \left| \det\left( \frac{\partial f_\theta(x,c)}{\partial x} \right) \right| dc \\
&= \int p_{x|c}(x|c) \cdot \left| \det\left( \frac{\partial f_\theta^{-1}(z,c)}{\partial z} \right) \right| \cdot \left| \det\left( \frac{\partial f_\theta(x,c)}{\partial x} \right) \right| dc \\
&= \int p_{x|c}(x|c)dc \\
&= p_x(x).
\end{aligned}
\tag{20}
$$

This indicates that with $c'$ consistent to $c$, we have $\hat{x} \sim p_x(x)$.

**Case (b)**: To represent inconsistency between $c'$ and $c$, we assume that $c'$ is randomly sampled, independent from $c$. The distribution $\hat{p}_x(x;\theta)$ is modified as:

$$
\begin{aligned}
\hat{p}_x(x;\theta) &= \int \hat{p}_{x|c'}(x|c';\theta)dc' \\
&= \int p_{z|c'}(f_\theta(x,c)|c') \cdot \left| \det\left( \frac{\partial f_\theta(x,c)}{\partial x} \right) \right| dc' \\
&= p_z(f_\theta(x,c)) \cdot \left| \det\left( \frac{\partial f_\theta(x,c)}{\partial x} \right) \right| \\
&= p_x(x) \cdot \left| \det\left( \frac{\partial f_\theta^{-1}(z,c')}{\partial z} \right) \right| \cdot \left| \det\left( \frac{\partial f_\theta(x,c)}{\partial x} \right) \right|.
\end{aligned}
\tag{21}
$$

Deriving from Equation (4), we have:

$$
\begin{aligned}
\left| \det\left( \frac{\partial f_\theta(x,c)}{\partial x} \right) \right| &= \begin{vmatrix} \frac{\partial z_1}{\partial x_1} & \frac{\partial z_1}{\partial x_2} \\ \frac{\partial z_2}{\partial x_1} & \frac{\partial z_2}{\partial x_2} \end{vmatrix} \\
&= \begin{vmatrix} 1 & \frac{\partial \alpha}{\partial x_2} \\ x_2 e^{\beta(z_1,c)} \frac{\partial \beta}{\partial z_1} + \frac{\partial \gamma}{\partial z_1} & e^{\beta(z_1,c)} + x_2 e^{\beta(z_1,c)} \frac{\partial \beta}{\partial z_1} \frac{\partial \alpha}{\partial x_2} + \frac{\partial \gamma}{\partial z_1} \frac{\partial \alpha}{\partial x_2} \end{vmatrix} \\
&= e^{\beta(z_1,c)}.
\end{aligned}
\tag{22}
$$

With Equation (5), we have:

$$
\left| \det\left( \frac{\partial f_\theta^{-1}(z,c')}{\partial z} \right) \right| = e^{-\beta(z_1,c')}.
\tag{23}
$$

Thus, $\hat{p}_x(x;\theta)$ can be further simplified to:

$$
\hat{p}_x(x;\theta) = p_x(x) \cdot e^{\beta(z_1,c)-\beta(z_1,c')},
\tag{24}
$$

where $\theta$ can be trained to make the distribution of $\hat{x}$ away from the distribution of $x$ when $c'$ is inconsistent with $c$.

## B. Structure of the Dynamic Generation Module

Traditional static keys pose a security risk, as the compromise of a single key could expose all secret images. To mitigate this, we introduce a dynamic generation strategy that dynamically generates lock-key pairs, ensuring that the lock and key for each secret image are unique. As a result, even if the key for one image is intercepted, the security of other images remains unaffected.

**In the forward hiding process**, as outlined in Table 4, we utilize several convolutional layers and a max-pooling layer to generate a global lock from the cover image and a local lock from the designated secret image. The fusion lock is then

*Table 4.* The detail structure of our proposed Lock-Key Generation Module, which is composed of 4 convolutional layers and an optional MaxPool layer.

| INPUT | $C_o$ $(H \times W \times 3)$ | $S_o^i$ $(H \times W \times 3)$ | $T$ $(H \times W \times 3)$ | $S_{rc}^i$ $(H/2 \times W/2 \times 3)$ |
|---|---|---|---|---|
| | CONV (3×3, 64) + RELU | | CONV (3×3, 64) + RELU | |
| | CONV (3×3, 64) + RELU | | CONV (3×3, 64) + RELU | |
| LAYERS | MAXPOOL (2×2) | | MAXPOOL (2×2) | - |
| | CONV (3×3, 64) + RELU | | CONV (3×3, 64) + RELU | |
| | CONV (3×3, 3) + RELU | | CONV (3×3, 3) + RELU | |
| OUTPUT | $L^i$ $(H/2 \times W/2 \times 3)$ | | $K^i$ $(H/2 \times W/2 \times 3)$ | |

*Table 5.* The results of hiding 6, 7, 8 secret images with our AIS method.

| N | DATASETS | COVER-STEGO | | | SECRET-REVEAL | | |
|---|---|---|---|---|---|---|---|
| | | PSNR↑ | SSIM↑ | LPIPS↓ | PSNR↑ | SSIM↑ | LPIPS↓ |
| 6 | DIV2K | 33.822 | 0.869 | 0.430 | 26.030 | 0.737 | 2.690 |
| | IMAGENET | 33.205 | 0.856 | 0.729 | 28.274 | 0.791 | 1.957 |
| 7 | DIV2K | 33.733 | 0.864 | 0.332 | 25.751 | 0.728 | 2.447 |
| | IMAGENET | 34.606 | 0.864 | 0.396 | 27.588 | 0.769 | 1.614 |
| 8 | DIV2K | 29.072 | 0.708 | 1.373 | 25.140 | 0.736 | 3.564 |
| | IMAGENET | 30.838 | 0.812 | 1.394 | 27.287 | 0.765 | 2.763 |

obtained through linear combination, formulated as: $L^i = \alpha \times f(C_o) + \beta \times f(S_o^i)$, where $f$ represents a learnable function, and $\alpha$, $\beta$ are user-defined coefficients.

**In the backward revealing process**, a similar structure is employed to generate the corresponding key from the stego image and the revealed compressed secret image, as described by: $K^i = \alpha \times g(T) + \beta \times g'(S_{rc}^i)$, where $\alpha$ and $\beta$ are shared with the forward process, and $g'$ and $g$ share the same parameters. However, the max-pooling layer is excluded in $g'$ since $S_{rc}$ is half the size of $T$. Both the global key and local key must match the corresponding lock to ensure that only the intended recipient can reconstruct the correct key.

## C. Training Details

The proposed Efficient and Separate Authentication Image Steganography Network is trained and tested on the DIV2K and ImageNet datasets. The DIV2K dataset consists of 800 training images cropped to 144×144 and 100 test images cropped to 1024×1024. For ImageNet dataset, we randomly select 20,000 training images cropped to 144×144 for finetuning and 5,000 test images cropped to 256×256. Training is performed for 100K iterations using the Adam optimizer with $\beta_1 = 0.9$ and $\beta_2 = 0.999$. The initial learning rates are set to $2 \times 10^{-4}$ for IAN, and $1 \times 10^{-4}$ for IHN and Dynamic Generation Module, with a CosineAnnealingLR scheduler for dynamic adjustment. The hyperparameters $\lambda_1$, $\lambda_2$ and $\lambda_3$ are set to 2, 4, and 3, respectively. All experiments are conducted on a Nvidia RTX 4090 GPU.

For the three baselines, ISN is reproduced based on the original paper, while DeepMIH and IIS are retrained using open-source codes. Performance is evaluated using Peak Signal-to-Noise Ratio (PSNR) and Structural Similarity (SSIM), where higher values indicate better quality, and Learned Perceptual Image Patch Similarity (LPIPS), where lower values suggest superior quality.

## D. More Experiments

### D.1. Very Large Capacity

To explore hiding with very large capacity, we conduct experiments on hiding more secret images with our proposed AIS. From the results shown in Table 5 and Figure 9, AIS achieves high performance even hiding 8 secret images in a single cover image. This indicates the superior effect of our proposed IAN, which integrates distribution adaptation and primary

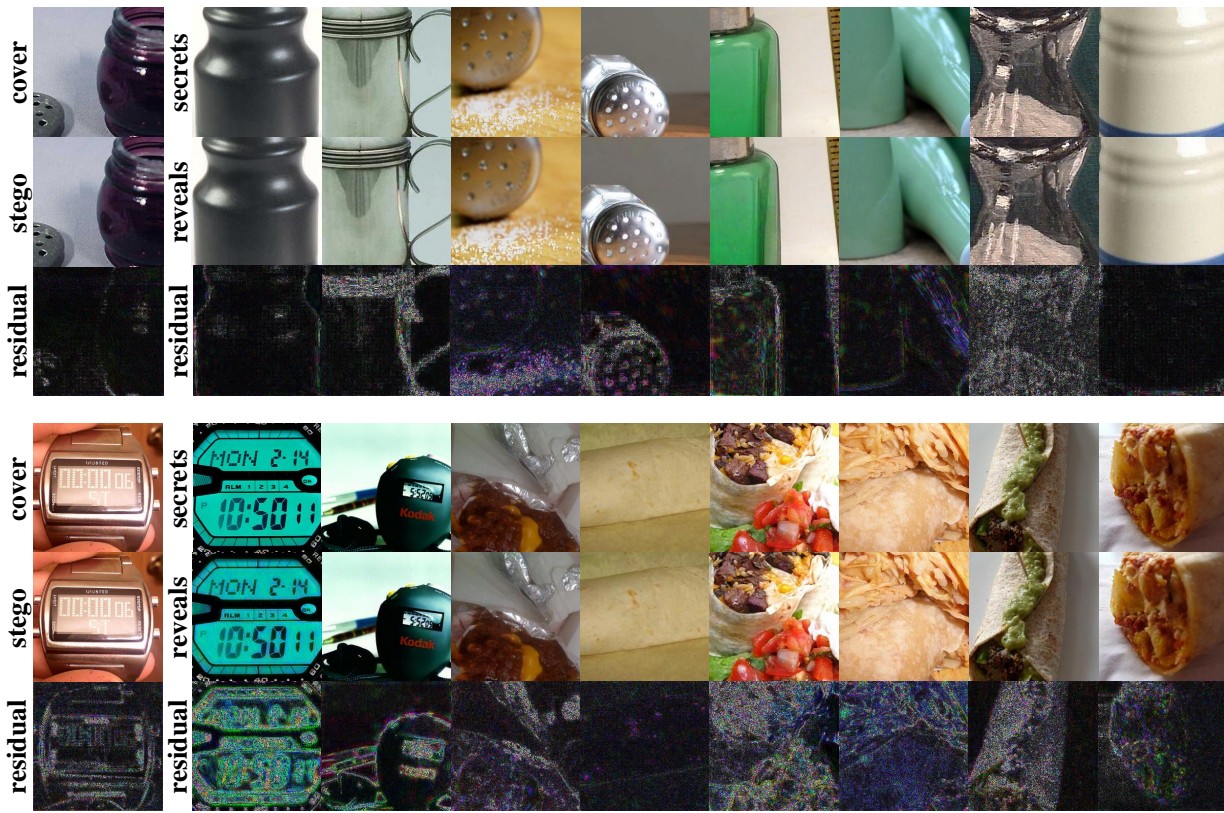

*Figure 9.* The visual results of hiding 8 secret images with our AIS method.

information extraction. Through embedding less authentication information and hiding only primary secret information, more secret images can be hidden and revealed in high quality.

### D.2. Visualization of Distribution Adaptation

To better demonstrate the effectiveness of distribution adaptation in the proposed AIS, we visualize the adapted primary information generated by IAN in Figure 10. For comparison, we also visualize the secret images and lock information in the traditional one-stage method. From the result, the distribution of secret images and lock information is significantly inconsistent. When attempting to embed both the secret information and the lock information, such consistency may occupy a large space, which reduces the quality of the stego images. Our proposed AIS makes distribution adaptation, integrating the secret information and lock information into a representation with consistent distribution. This may release some space for hiding more information to some extent.

### D.3. Model of Redundancy Prediction Module

In the forward process of IHN, redundant information is generated, which can not be transmitted through the public channel. Therefore, auxiliary redundancy is required to reveal the secret information. Traditional methods (Jing et al., 2021; Lu et al., 2021; Guan et al., 2022) randomly samples a case-agnostic variable from the Guassian distribution, which is inconsistent with the distribution of the original redundant information. This weakens reversibility of the network to some extent. The Redundancy Prediction Module is employed to learn case-specific auxiliary redundancy with a consistent distribution. IIS (Zhou et al., 2024) has proved the effectiveness of residual blocks on this target, for the residual structure can learn deep features.

We conduct experiments on replacing the residual blocks with other model structure, such as UNet, Vision Transformer

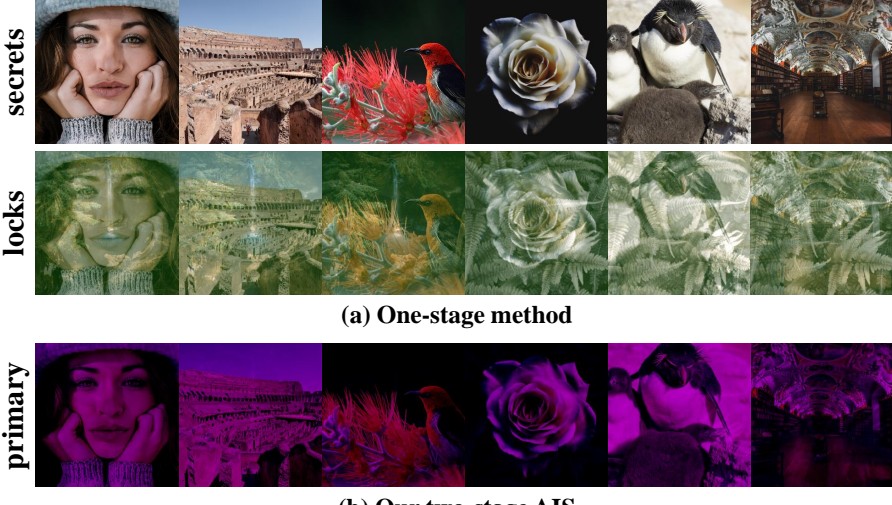

(a) One-stage method

(b) Our two-stage AIS

*Figure 10.* The visual comparison between secret and lock information of other methods and distribution adapted primary information of our AIS.

*Table 6.* Comparison of different model structure as Redundancy Prediction Module.

| MODEL | PARAMS | COVER-STEGO | | | SECRET-REVEAL | | |
| --- | --- | --- | --- | --- | --- | --- | --- |
| | | PSNR↑ | SSIM↑ | LPIPS↓ | PSNR↑ | SSIM↑ | LPIPS↓ |
| UNET | 1.933M | 34.039 | 0.898 | 0.821 | **35.036** | **0.909** | 0.666 |
| VIT | 129.8M | 35.758 | **0.920** | 0.414 | 34.318 | 0.879 | 0.498 |
| CBAM | 0.001M | **35.762** | 0.918 | 0.440 | 34.546 | 0.888 | 0.540 |
| GUASSIAN SAMPLING | 0 | 29.229 | 0.81 | 1.499 | 30.516 | 0.819 | 1.049 |
| **RESBLOCK(OURS)** | 0.013M | 34.721 | 0.904 | **0.396** | 34.761 | 0.903 | **0.420** |

(ViT), Convolutional Block Attention Module (CBAM). Considering the lightweight design of the module, we use a simple form of UNet with only a single downsampling and upsampling. ViT is a simplified implement with only one layer. An additional convolutional layer is added to CBAM to match the number of output channels. We also include the traditional Gaussian sampling strategy in the experiments. From the results shown in Table 6, residual blocks generate stego and revealed secret images with the best LPIPS. Compared to other model structure, residual blocks achieve a more balanced performance between stego images and revealed secret images, while keeps the model size small.

### D.4. Resistance against Real Scene Disturbance

We have conducted experiment on DIV2K dataset, hiding 3 secret images, under the perturbations of Guassian noise, Poisson noise, and JPEG compression, respectively. We visualize some samples of revealed secret images under different perturbation scales. From the results shown in Figure 11, our proposed AIS can withstand noise within a certain range. When the $\sigma$ of Guassian noise is set to 1 and the scale of Poisson noise is set to 1, the revealed secret image is almost not affected. When the scale grows, contour information is still preserved, though noise can be observed visually. For JPEG compression, our method can still reveal the contour information of the secret images, while the color distortion and detail information loss are severe.

The result is mainly because our proposed AIS focus on the security of authentication mechanism. This requires that when the key is different from the lock, the revealed information is far from the original secret images. Through such a target, the Invertible Authentication Network is trained to be sensitive to minor disturbances. This guarantees that the key should be as similar as possible to the corresponding lock. Otherwise, random information is introduced to the revealed information to cover the secret information by the network. Sensitivity makes a trade-off between security and robustness. While the authentication mechanism is effective, it also makes the network sensitive to real-world scene disturbances.

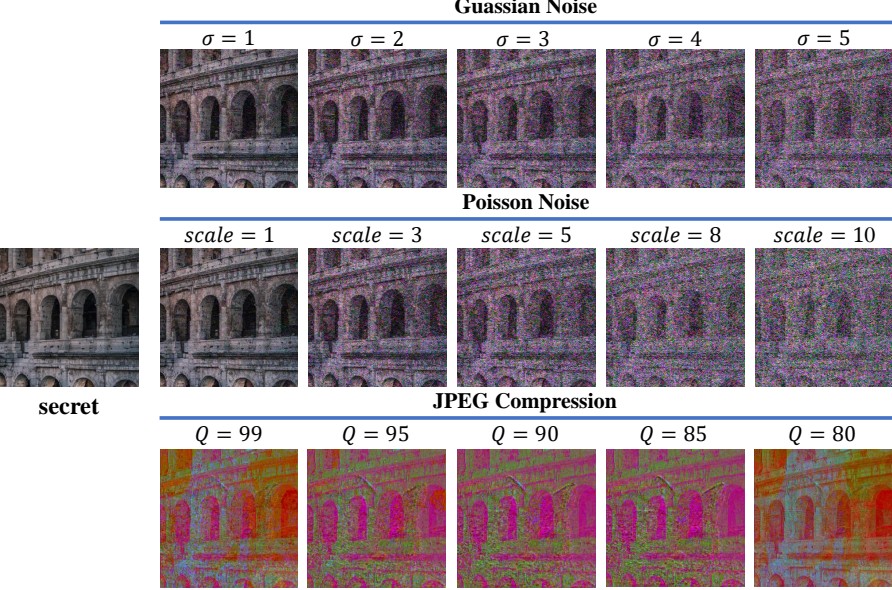

*Figure 11.* The visual result of resistance against real-world scene disturbance.

In future work, for better robustness, we suggest adding an enhancement module to enhance the key information under JPEG compression, as well as a noise-guided module to reduce the disturbance of specific noise. Such a method has been partly researched in (Xu et al., 2022) and can be applied to our proposed AIS. To balance robustness and security, an additional attention block may need to be added to focus more on the difference between the key and the lock to catch slight changes.

