# OpenReview forum: "Efficient and Separate Authentication Image Steganography Network"
_ICML.cc/2025/Conference — ICML 2025 spotlightposter_

### Official Review · Reviewer_JeET · 2025-03-10

**Overall Recommendation:** 4

**Summary:**

This paper presents a novel authentication-based image steganography framework (AIS) with separate invertible networks for authentication and hiding. The approach addresses critical challenges in multi-recipient security and large-capacity hiding. The experiments demonstrate significant improvements in stego/revealed image quality and computational efficiency.

**Claims And Evidence:**

Yes, they are clear and have convincing evidence.

**Essential References Not Discussed:**

N/A

**Experimental Designs Or Analyses:**

All the experimental designs or analyses have been checked.

**Methods And Evaluation Criteria:**

Yes, they are promising.

**Other Comments Or Suggestions:**

None.

**Other Strengths And Weaknesses:**

The paper contains the following weaknesses:
1) The comparison in Table 1 shows superior performance over baselines. However, recent diffusion-based steganography methods are not included. Although they are not in the same class, could the authors discuss how your method compares to diffusion models in terms of security and computational overhead?
2) The dynamic lock-key generation module is critical. Could you clarify how the "linear combination" of global and local locks ensures uniqueness across multiple secret images?
3) The paper claims parallel hiding reduces model size, but Table 1 shows AIS still requires ~6M parameters for N=5. Could you provide FLOPs/parameter ratios relative to baselines to better illustrate efficiency gains?
4) Some formatting inconsistencies exist. Please proofread for hyphenation and notation consistency.

**Questions For Authors:**

1) The security analysis evaluates resistance to SRNet and ZhuNet. Have you tested against more advanced steganalysis tools to further validate robustness?
2) The IAN extracts "primary information" from secret images. How is the 3-channel design justified compared to other feature reduction strategies (e.g., PCA or attention-based compression)?

**Relation To Broader Scientific Literature:**

The paper addresses critical challenges in multi-recipient security and large-capacity hiding. This will have a broader study topics such as multi-recipient security, large-capacity hiding, video hiding.

**Theoretical Claims:**

All the proofs for theoretical claims have been checked.

---

> ### Author Rebuttal · Authors · 2025-03-30
>
> Thank you very much for your valuable comments. We hope that the explanations on the questions can help you better understand our proposed method.
>
> Q1: The comparison in Table 1 shows superior performance over baselines. However, recent diffusion-based steganography methods are not included. Although they are not in the same class, could the authors discuss how your method compares to diffusion models in terms of security and computational overhead?
>
> A1: We compare our method with diffusion-based models. (1) Capacity. Diffusion-based models can only hide 1 secret image once, which limits flexibility. AIS can hide up to 8 secret images in a cover image. (2) Quality. The average PSNR of revealed secret images of diffusion-based models is typically below 25. AIS achieves PSNR of over 25 even when hiding 8 secret images. (3) Security. Diffusion-based models ignore the problem that keys cannot be transmitted through public channel. AIS can restore keys through Lock-Key Generation Module, ensuring security of keys. (4) Computational overhead. Diffusion-based models typically require billions of parameters and a long inference time to generate images. AIS only requires millions of parameters and time below 0.2s. These comparisons indicate that AIS performs better in large-capacity steganography.
>
> Q2: The dynamic lock-key generation module is critical. Could you clarify how the "linear combination" of global and local locks ensures uniqueness across multiple secret images?
>
> A2: Through extracting multiscale features of cover and secret images, Dynamic Lock-Key Generation Module generates a global lock from cover image and a local lock from each secret image. This means within the same batch, global lock is the same, while local lock for each secret image is different from each other. For different batches, global lock is different from each other. Linear combination integrates global locks and local locks with two coefficients to control the strength of the two lock components. Higher coefficient before global locks focuses on the uniqueness among different batches. Higher coefficient before local locks enhances uniqueness among different secret images within the same batch.
>
> Q3: The paper claims parallel hiding reduces model size, but Table 1 shows AIS still requires ~6M parameters for N=5. Could you provide FLOPs/parameter ratios relative to baselines to better illustrate efficiency gains?
>
> A3: We have compared FLOPs and inference time in the original paper, which are shown in Table 1, Column FLOPs and Time. For FLOPs, AIS has the least FLOPs among all the baselines. For inference time, AIS spends shorter inference time compared to traditional serial hiding methods. The results indicate that AIS achieves a better balance between quality and efficiency.
>
> Q4: Some formatting inconsistencies exist. Please proofread for hyphenation and notation consistency.
>
> A4: We check the paper again and make some correction. (1) “CIS” in Table 2 is modified to “AIS” for consistency. (2) The font of “exp” in Equation 5 and 6 is modified to be consistent with Equation 4 and 7. (3) “low-” in page 4 is modified to “low-frequency” for correct hyphenation. (4) “$T$” in page 5 is modified to “$T_f$” to correctly denote the frequency stego image. Other grammar issues are also solved in the revised paper.
>
> Q5: The security analysis evaluates resistance to SRNet and ZhuNet. Have you tested against more advanced steganalysis tools to further validate robustness?
>
> A5: We have added experiments on resistance against LWENet[1], a more advanced steganalysis method. The detection accuracy is 67.30%(ISN), 89.55%(DeepMIH), 62.20%(IIS) and 51.75%(our AIS). This means the stego images generated by AIS confuse the steganalysis model, which improves security against the attackers. The results will be added to the paper.
>
> [1] Weng S, Chen M, Yu L, Sun S. Lightweight and effective deep image steganalysis network. IEEE Signal Processing Letters, 29:1888-92, 2022.
>
> Q6: The IAN extracts "primary information" from secret images. How is the 3-channel design justified compared to other feature reduction strategies (e.g., PCA or attention-based compression)?
>
> A6: (1) The design of reversibility in IAN can reveal the original secret image without extra modules. In the backward process, the original secret images can be revealed from the primary information through the reverse process of IAN with shared parameters. In contrast, other strategies can only extract critical information. They require additional modules to reveal the original secret images. (2) Invertible Neural Network implies a prior that the data distribution can be decomposed into a simple distribution through learnable transformations. This enables IAN to model the other information into a simple distribution. Through a prior of Gaussian distribution, part of the other information can be restored. This improves the quality of revealed secret images, making IAN suitable for reconstruction tasks.

---

> > ### Comment · Reviewer_JeET · 2025-04-02
> >
> > Thanks for the author's reply, which solved most of my problems. I will consider raising the score appropriately.

---

> > > ### Author Response · Authors · 2025-04-08
> > >
> > > We greatly appreciate your acknowledge of our work and raising the score. Thanks again for your time and effort in reviewing our paper.

---

### Official Review · Reviewer_sPRs · 2025-03-10

**Overall Recommendation:** 3

**Summary:**

This work advances adaptive steganography by redefining secure multi-user communication through the integration of invertible networks and dynamic authentication. A key innovation lies in decoupling authentication from secret data via distribution adaptation, a conceptual leap that reimagines information hiding in constrained spaces. The parallel hiding architecture further exemplifies strong engineering intuition, achieving parameter reduction without compromising quality. Beyond its forward-looking contributions, the paper also addresses a pressing challenge in privacy-preserving multi-user steganography.

**Claims And Evidence:**

Yes. The claims are well-supported with clear evidence.

**Essential References Not Discussed:**

Yes, the cited works sufficiently contextualize the paper’s key contributions.

**Experimental Designs Or Analyses:**

Yes, I have checked all the experimental designs or analyses.

**Methods And Evaluation Criteria:**

Yes. The methods and evaluation criteria are appropriate.

**Other Comments Or Suggestions:**

See "Other Strengths And Weaknesses".

**Other Strengths And Weaknesses:**

Several aspects of the paper could be improved:
1. The dynamic lock-key module employs a "simplified UNet." Providing ablation studies on alternative architectures or clarifying the rationale for this choice would strengthen the justification.
2. In Figure 3, clearer annotations for the lock/key propagation paths would enhance the workflow diagram’s clarity.
3. The term "primary information extraction" requires a more precise and well-supported explanation.
4. The proposed method could potentially be extended to video steganography. Could the authors suggest how to transfer the adaptation strategy to video steganography with temporal consistency requirements?
5. Minor grammar issues should be addressed for improved readability.

**Questions For Authors:**

See "Other Strengths And Weaknesses".

**Relation To Broader Scientific Literature:**

The methods in this paper can well address the problems within the practical engineer, and gives a new direction for multi-user privacy security in this area.

**Theoretical Claims:**

Yes, I have checked all the proofs.

---

> ### Author Rebuttal · Authors · 2025-03-30
>
> Thank you very much for your valuable comments. We hope that the explanations on the questions can help you better understand our proposed method.
>
> Q1: The dynamic lock-key module employs a "simplified UNet." Providing ablation studies on alternative architectures or clarifying the rationale for this choice would strengthen the justification.
>
> A1: Dynamic Lock-Key Generation Module is employed to extract features from the cover image and each secret image to dynamically generate locks and keys. By feature extraction, the locks for secret images are different from each other, ensuring security among multiple recipients. Many traditional models of extracting features can achieve this. In AIS, we employ a simplified UNet due to two reasons. (1) Multiscale feature fusion. UNet can extract features of different scales through symmetric downsampling-upsampling structure and skip connection. It can extract detailed information such as edges and textures in shallow layers, and capture abstract semantics in deep layers. This makes the generated locks and keys have richer representation, thus improving security. (2) Lightweight. The simplified UNet requires only 77K parameters and 0.395G FLOPs, ensuring efficiency.
>
> Q2: In Figure 3, clearer annotations for the lock/key propagation paths would enhance the workflow diagram’s clarity.
>
> A2: We have modified Figure 3 (https://s1.imagehub.cc/images/2025/03/26/d4d07474bef137e5714acdbd5c45b36b.png). The annotations of output lock and key have been added. Additionally, the paths of inputs have been added. Locks are generated from cover and secret images. Keys are generated from stego images and revealed primary information.
>
> Q3: The term "primary information extraction" requires a more precise and well-supported explanation.
>
> A3: Primary information is the part of information that is more important for restoring the original image. On the one hand, the space of a cover image for hiding secret is limited. With more information hidden in the cover image, the quality of the generated stego image will be lower. On the other hand, sufficient information needs to be hidden in the cover image for revealing. With less information hidden in the cover image, the quality of revealed secret images will be lower. The steganography network is optimized to achieve a balance between the quality of stego and revealed secret images. Based on this, a spontaneous idea is to hide part of the secret information. Xiao et al.[1] pointed out that for a 12-channel frequency domain RGB image, the low-frequency information of 3 channels is sufficient to restore the original image of high quality. Motivated by this, we set the number of reserved channels as 3. By jointly optimizing stego images and revealed secret images, IAN is trained to extract information which is more important for revealing. We call this more primary information. This part of information can enhance revealing while hiding less information in the cover image, improving the quality of both stego and revealed secret images, which meets the goal of joint optimization.
>
> [1] Xiao, M., Zheng, S., Liu,C., Lin,Z., and Liu,T. Invertible rescaling network and its extensions. International Journal of Computer Vision, 131(1): 134–159, 2023.
>
> Q4: The proposed method could potentially be extended to video steganography. Could the authors suggest how to transfer the adaptation strategy to video steganography with temporal consistency requirements?
>
> A4: It’s a good idea to apply distribution adaptation to video steganography. When hiding secret information into a cover video, inconsistent distribution of the secret information may cause visual discontinuity between adjacent frames, which degrades temporal consistency. We suggest a module similar to IAN being introduced to video steganography. This module receives the input of secret information, along with a cover video as the condition. With similar transformation, the secret information is decomposed into more primary information with consistent distribution. At the same time, the transformation may pay attention to the cover video condition, and guide the generation of the primary information. With more primary and consistent information hidden, the stego video may release some space for hiding more information, while keeps consistent distribution between frames.
>
> Q5: Minor grammar issues should be addressed for improved readability.
>
> A5: Thank you for your suggestion. We check the paper again and make some correction. (1) “requirement” in page 1 is modified to “requirements”. (2) “CIS” in Table 2 is modified to “AIS” for consistency. (3) The font of “exp” in Equation 5 and 6 is modified to be consistent with Equation 4 and 7. (4) “low-” in page 4 is modified to “low-frequency” for correct hyphenation. (5) “$T$” in page 5 is modified to “$T_f$” to correctly denote the frequency stego image. In addition, other grammar issues have also been solved in the revised paper.

---

> > ### Comment · Reviewer_sPRs · 2025-04-07
> >
> > Thank you for detailed reply. My concerns have been addressed, and I will maintain my current score.

---

> > > ### Author Response · Authors · 2025-04-08
> > >
> > > We greatly appreciate your acknowledge of our work and respect your decision on maintaining the score. Thanks again for your time and effort in reviewing our paper.

---

### Official Review · Reviewer_Afwk · 2025-03-12

**Overall Recommendation:** 3

**Summary:**

The paper presents AIS, an authentication-based steganography framework that leverages invertible networks for distribution adaptation and parallel hiding. The two-stage design and focus on security and efficiency are commendable, and comprehensive experiments demonstrate the framework’s advantages. While the paper is well-structured and the core contributions are promising, a deeper analysis of key design choices and broader applicability would strengthen the work’s overall impact.

**Claims And Evidence:**

Yes, they are supported.

**Essential References Not Discussed:**

The related works are essential for understanding the key contributions of the paper and are appropriately cited and discussed.

**Experimental Designs Or Analyses:**

I have reviewed the soundness and validity of the experimental designs and analyses, and they are accurate.

**Methods And Evaluation Criteria:**

Yes, they make sense and are widely used in this area.

**Other Comments Or Suggestions:**

Although the paper is well-structured and presents some new ideas, there are several issues that need clarification and resolution (refer to the weaknesses). If these concerns are addressed in the rebuttal phase, I may reconsider my decision.

**Other Strengths And Weaknesses:**

Strengths: The proposed AIS framework elegantly integrates authentication with invertible networks, achieving secure and efficient multi-image steganography. The two-stage design and distribution adaptation strategy are innovative, contributing to improved performance and robustness. The paper is well-structured and presents comprehensive experimental results.

Weaknesses:
a) The distribution adaptation mechanism is central to reducing lock-induced artifacts. Could the latent distributions of locks and secrets before and after adaptation be visualized to quantitatively demonstrate alignment?
b) While Table 5 (Appendix D) reports results for N = 8, the main text focuses on N = 2–5. Including results for N = 6–7 could provide insight into performance degradation trends.
c) The RPM module in IHN uses a residual structure for redundancy prediction. How does this compare to alternative prediction architectures, such as transformers? Ablation studies on RPM’s design would strengthen the claims.
d) The invertible blocks (Eq. 4–7) share parameters between forward and backward passes. Does this limit flexibility compared to non-shared designs? A discussion of the trade-offs would clarify the design rationale.
e) Some equations have formatting issues. A thorough review and correction are recommended.
f) How does AIS handle real-world scenarios such as JPEG compression or noise perturbations? Testing under such conditions would better demonstrate the framework's practicality.

**Questions For Authors:**

Please refer to the weaknesses.

**Relation To Broader Scientific Literature:**

The method relates to authentication-based steganography designed for multiple users.

**Theoretical Claims:**

I have reviewed the proof of Theorem 14 in the appendix, and it appears to be correct.

---

> ### Author Rebuttal · Authors · 2025-03-30
>
> Thank you very much for your valuable comments. We hope that the explanations on the questions can help you better understand our proposed method.
>
> Q1: The distribution adaptation mechanism is central to reducing lock-induced artifacts. Could the latent distributions of locks and secrets before and after adaptation be visualized to quantitatively demonstrate alignment?
>
> A1: Figure 7(b) shows mean and standard deviation of 1000 samples to generally indicate consistent distribution of adapted information. According to your suggestion, we visualize some locks, secret images and adapted information (https://s1.imagehub.cc/images/2025/03/26/edfaea8efa039b0ebef270cecd585a39.png). From the result, the distribution of secret images and locks is significantly inconsistent. AIS makes distribution adaptation to get a representation with consistent distribution. The above content will be added to the paper.
>
> Q2: While Table 5 (Appendix D) reports results for N = 8, the main text focuses on N = 2–5. Including results for N = 6–7 could provide insight into performance degradation trends.
>
> A2: We show results of N=8 to indicate AIS’s superior performance in large capacity. To highlight very large capacity, we didn’t show results of N=6,7. On DIV2K dataset, hiding 6 secret images, cover-stego achieves (33.822, 0.869, 0.430) of (PSNR, SSIM, LPIPS), and secret-reveal achieves (26.030, 0.737, 2.690). When hiding 7 secret images, results are (33.733, 0.864, 0.332) and (25.751, 0.728, 2.447). Following your suggestion, the complete results will be added to the paper.
>
> Q3: The RPM module in IHN uses a residual structure for redundancy prediction. How does this compare to alternative prediction architectures, such as transformers? Ablation studies on RPM’s design would strengthen the claims.
>
> A3: RPM is employed to learn an auxiliary variable with consistent distribution to redundancy, which improves reversibility. Many traditional models of extracting features can achieve this. Residual structure has fewer parameters and can extract deeper features. Following your suggestion, we conduct experiments on structures of UNet, Vision Transformer(ViT), Convolutional Block Attention Module(CBAM) and traditional Gaussian sampling strategy (https://s1.imagehub.cc/images/2025/03/27/3c23350577d0befe5b36a40be11b2a32.jpeg). Compared to UNet and ViT, residual blocks achieve a balance between quality and model size. Compared to CBAM, residual blocks reveal secret images of higher quality. The above content will be added to the paper.
>
> Q4: The invertible blocks (Eq. 4–7) share parameters between forward and backward passes. Does this limit flexibility compared to non-shared designs? A discussion of the trade-offs would clarify the design rationale.
>
> A4: For image steganography task, non-shared design will not improve flexibility. Hiding and revealing are a pair of reversible processes. In forward process, network receives cover and secret images, and outputs a stego image and redundancy. In backward process, stego image and auxiliary variable are input to get the revealed secret images. Through RPM, auxiliary variable has a consistent distribution with redundancy. In this case, shared parameters can ensure one-to-one correspondence between input and output, improving reversibility and quality. In contrast, non-shared design has disadvantages of (1) double number of parameters. (2) poor reconstruction quality due to accumulation of errors. (3) unstable training due to higher complexity.
>
> Q5: Some equations have formatting issues. A thorough review and correction are recommended.
>
> A5: We check the paper again and fix some formatting and grammar issues. (1) Wrong format of exp in Equation 5,6. (2) Wrong punctuation in Equation 12. (3) Wrong punctuation in Equation 24. (4) Word error “CIS” in caption of Table 2. Other grammar issues have also been solved.
>
> Q6: How does AIS handle real-world scenarios such as JPEG compression or noise perturbations? Testing under such conditions would better demonstrate the framework's practicality.
>
> A6: Robustness is another important task in the field of image steganography, aiming to reduce interference of real-world noise and compression. Experiments of such conditions are typically conducted under this task. AIS focuses on security and capacity, with a different target from robustness. Due to this target, AIS is sensitive to minor changes of keys to guarantee similarity of keys and locks. Sensitivity makes a trade-off between security and robustness. While the authentication mechanism is effective, it also makes AIS sensitive to disturbance. It is a good suggestion to consider robustness in large-capacity image steganography with authentication. In future work, we suggest adding an enhancement module to resist compression, and a noise-guided module to reduce disturbance of noises. An attention block may be added to catch slight changes for security. The above contents will be added in new version of the paper.

---

### Official Review · Reviewer_Y3wr · 2025-03-12

**Overall Recommendation:** 4

**Summary:**

This paper makes a substantial contribution to secure multi-recipient image steganography by addressing the critical yet under explored challenge of authentication. The proposed AIS framework elegantly integrates authentication through a novel two-stage architecture—IAN for lock-key generation and distribution adaptation, and IHN for parallelized hiding—demonstrating rigorous theoretical grounding in normalizing flows. The authors provide compelling empirical evidence, including JS divergence analysis and PSNR/SSIM metrics across DIV2K datasets, to validate their claim that distribution adaptation mitigates spatial conflicts between authentication locks and secret images. Particularly noteworthy is the method’s ability to reduce model parameters, which addresses a longstanding efficiency-quality trade-off in high-capacity steganography. The discussion of dynamic lock-key generation’s resistance to brute-force attacks adds practical security insights. This research may set a promising foundation for authentication-aware steganography, but still with some minor clarifications.

**Claims And Evidence:**

It is clear and convincing.

**Essential References Not Discussed:**

The references are enough.

**Experimental Designs Or Analyses:**

Yes, they are reasonable.

**Methods And Evaluation Criteria:**

Yes.

**Other Comments Or Suggestions:**

Please see the questions.

**Other Strengths And Weaknesses:**

Please see the questions.

**Questions For Authors:**

1. Could the authors clarify whether the dynamic lock-key generation strategy considers potential adversarial attacks on key generation?
2. The JS divergence analysis in Figure 2 is insightful. Would including KL divergence metrics further strengthen the distribution comparison?
3. In Section 3.1, the mathematical formulation of authentication feasibility could benefit from expanded derivations for reproducibility.
4. Could runtime metrics (e.g., inference speed) be added to assess practical efficiency?

**Relation To Broader Scientific Literature:**

This paper provides a promising approach for independence recovery for multiple image steganography.

**Theoretical Claims:**

Correct.

---

> ### Author Rebuttal · Authors · 2025-03-30
>
> Thank you very much for your valuable comments. We hope that the explanations on the questions can help you better understand our proposed method.
>
> Q1: Could the authors clarify whether the dynamic lock-key generation strategy considers potential adversarial attacks on key generation?
>
> A1: The security of locks and keys is ensured through several special designs. (1) Dynamic generated rather than static. In AIS, we generate different pairs of locks and keys for each secret image with a trainable Dynamic Generation Module. These pairs are generated through feature fusion between cover images and corresponding secret images. Unlike traditional static key, our method ensures specific locks and keys for each recipient, thereby mitigating adversarial attack risks through reduced key exposure predictability. (2) Generated by trainable networks rather than specific cryptographic patterns. Traditional cryptographic methods rely on a predefined pattern, which is easy to be attacked by cryptanalysis techniques. In AIS, we train a lightweight network with linear and nonlinear transformations. This approach establishes security dependence on network parameter confidentiality, which is hard to crack. (3) Two stage revealing. In the revealing process, the recipient should first reveal primary information with IHN. Then a key is generated for authentication and reconstructing the original secret images from the primary information. This indicates that attackers should obtain both the primary information and the key to fetch the secret images. Since IHN is a more complex Invertible Neural Network, it can be very hard for effective adversarial attacks.
>
> Q2: The JS divergence analysis in Figure 2 is insightful. Would including KL divergence metrics further strengthen the distribution comparison?
>
> A2: In Figure 2, we calculate JS divergence to indicate the distribution differences between cover and stego images of authentication-based and authentication-free methods. JS divergence is calculated by $JS(p,q)=1/2* KL(p,q)+1/2*KL(q,p)$, which includes KL divergence and is capable of indicating the distribution differences. According to your suggestion, we also calculate KL divergence. For authentication-free method, the value is 0.0444. For authentication-based method, the value is 0.0697. Since higher KL divergence indicates greater inconsistency between the two kinds of distribution, the result implies that stego images generated in authentication-based method may contain more information with inconsistent distribution, resulting in low quality. This is consistent with our discovery in the paper.
>
> Q3: In Section 3.1, the mathematical formulation of authentication feasibility could benefit from expanded derivations for reproducibility.
>
> A3: The expanded derivations are in Appendix A. We have provided necessary derivation processes and explanations. In the proof, $x$ denotes data domain of secret images. $z$ denotes latent domain after the secret images transformed by an invertible network flow, denoted as $f$. $p(x)$, $\hat{p}(x)$, $p(z)$ denote the distribution of secret images, revealed secret images and latent information, respectively. $c$ and $c’$ denotes lock and key, respectively. Our goal is to prove that when the distribution of $c$ and $c’$ are consistent, $\hat{p}(x)$ is consistent to $p(x)$. When $c$ and $c’$ are inconsistent, $\hat{p}(x)$ has to be away from $p(x)$. This ensures revealing of meaningless information when the key is wrong. Equation 1 is a general form of the change-of-variable formula. This indicates the calculation of $\hat{p}(x)$ through $p(z)$. $det(·)$ denotes the Jacobian determinant. Equation 2 is the log likelihood loss derived from Maximum Likelihood Estimation (MLE). We give the detailed derivation in Appendix A.1. Equation 3 is an extended form of Equation 1, adding the lock and key. It can be simply inferred from the conditional distribution. Based on Equation 3, we prove that $\hat{p}(x)=p(x)\cdot e^{β(z, c)-β(z, c’)}$ by expanding the Jacobian determinant, where $\beta$ is a trainable network. The derivation process is detailed in Appendix A.2. This means that the network can be trained to make $\hat{p}(x)$ away from $p(x)$ when $c$ and $c’$ are inconsistent, which is in line with our goal.
>
> Q4: Could runtime metrics (e.g., inference speed) be added to assess practical efficiency?
>
> A4: We have compared the inference time in the original paper, which is shown in Table 1, Column Time. From the result, AIS spends shorter inference time compared to traditional serial hiding methods. Especially when hiding 5 images, AIS only requires 121.1ms, shorter than DeepMIH(269.1ms) and IIS(439.6ms). Though ISN achieves inference time of 51.7ms, the hiding quality(30.522) and revealing quality(29.077) are lower than AIS(32.767 and 30.060). The results indicate that AIS achieves a better balance between quality and inference speed.

---

### Decision · Program_Chairs · 2025-05-01

**Decision:**

Accept (spotlight poster)

**Comment:**

To improve effectiveness and efficiency in large-capacity hiding, this paper proposes an image steganography network collaborating with separate authentication and efficient scheme, and demonstrates the authentication feasibility within image steganography. All reviewers give 'weak accept' or 'accept' scores. The idea of the paper is interesting. And the paper is well-structured and presents comprehensive experimental results.